# Exploring the Impact of Efavirenz on Aflatoxin B1 Metabolism: Insights from a Physiologically Based Pharmacokinetic Model and a Human Liver Microsome Study

**DOI:** 10.3390/toxins16060259

**Published:** 2024-06-04

**Authors:** Orphélie Lootens, Marthe De Boevre, Elke Gasthuys, Sarah De Saeger, Jan Van Bocxlaer, An Vermeulen

**Affiliations:** 1Centre of Excellence in Mycotoxicology and Public Health, Department of Bioanalysis, Ghent University, Ottergemsesteenweg 460, 9000 Ghent, Belgium; 2Laboratory of Medical Biochemistry and Clinical Analysis, Department of Bioanalysis, Ghent University, Ottergemsesteenweg 460, 9000 Ghent, Belgium; 3MYTOX-SOUTH^®^, International Thematic Network, Ghent University, Ottergemsesteenweg 460, 9000 Ghent, Belgium; 4Department of Biotechnology and Food Technology, Faculty of Science, University of Johannesburg, Doornfontein Campus, P.O. Box 17011, Gauteng 2028, South Africa

**Keywords:** mycotoxins, CYP450 enzymes, pharmacokinetics, in vitro, efavirenz, LC-MS/MS, aflatoxin B1, human liver microsomes

## Abstract

Physiologically based pharmacokinetic (PBPK) models were utilized to investigate potential interactions between aflatoxin B1 (AFB1) and efavirenz (EFV), a non-nucleoside reverse transcriptase inhibitor drug and inducer of several CYP enzymes, including CYP3A4. PBPK simulations were conducted in a North European Caucasian and Black South African population, considering different dosing scenarios. The simulations predicted the impact of EFV on AFB1 metabolism via CYP3A4 and CYP1A2. In vitro experiments using human liver microsomes (HLM) were performed to verify the PBPK predictions for both single- and multiple-dose exposures to EFV. Results showed no significant difference in the formation of AFB1 metabolites when combined with EFV (0.15 µM) compared to AFB1 alone. However, exposure to 5 µM of EFV, mimicking chronic exposure, resulted in increased CYP3A4 activity, affecting metabolite formation. While co-incubation with EFV reduced the formation of certain AFB1 metabolites, other outcomes varied and could not be fully attributed to CYP3A4 induction. Overall, this study provides evidence that EFV, and potentially other CYP1A2/CYP3A4 perpetrators, can impact AFB1 metabolism, leading to altered exposure to toxic metabolites. The results emphasize the importance of considering drug interactions when assessing the risks associated with mycotoxin exposure in individuals undergoing HIV therapy in a European and African context.

## 1. Introduction

Mycotoxins are secondary metabolites produced by fungi that contaminate food commodities, endangering food safety and, consequently, food security. Due to climatic, sociological, and agronomical factors and a lack of controlling mitigation strategies, some regions in the global South—e.g., Sub-Saharan Africa—are more affected than others [1]. High humidity levels promote the growth of fungi, while drought encourages the production of mycotoxins. The production of mycotoxins occurs at different stages, but the most extensive contamination is typically observed during harvest and storage, primarily caused by inadequate sanitation practices and improper food handling. Aflatoxin B1 (AFB1), a mycotoxin produced prominently by *Aspergillus flavus*, is carcinogenic to humans, immunotoxic, and hepatotoxic and classified into Group 1 by the International Agency for Research on Cancer (IARC). AFB1 can be found in various food commodities such as rice, cereals, spices, maize, peanuts, dried fruits and seeds [2]. AFB1 is considered to be an abundant mycotoxin in Sub-Saharan Africa. Exposure to aflatoxins can occur not only through the direct consumption of contaminated foods but also by milk consumption contaminated with aflatoxin M1 (AFM1) from animals that have been exposed to AFB1 through the ingestion of contaminated feed. Furthermore, the global impact of climate change on mycotoxin occurrence cannot be underestimated [3].

In humans, AFB1 is mainly metabolized in the liver by hepatic cytochrome P450 (CYP) enzymes [4,5]. Some metabolites are considered to be detoxifying, namely aflatoxin Q1 (AFQ1), which is considered to be 18 times less toxic and 83 times less mutagenic than AFB1, aflatoxin P1 (AFP1), and aflatoxin B2a (AFB2a). Another metabolite is aflatoxin M1 (AFM1), which is ten times less toxic than AFB1 [3,4,5,6,7,8,9,10]. AFM1 is the metabolite excreted in human and animal milk [11]. Other metabolites exert more toxicity than the parent compound, i.e., aflatoxin-8,9-endo/exo-epoxides (AFBO), by forming DNA adducts. The exo-epoxide is produced by CYP3A4 and CYP1A2 and has a higher affinity for guanine residues than the endo-epoxide, which is produced by CYP1A2. Exo-epoxide is also produced more in the liver; therefore, it is considered the main carcinogenic metabolite of AFB1 [7,12]. Other metabolites with the capacity to form adducts with DNA are AFM1, aflatoxicol H1 (AFH1), and aflatoxicol (AFL). An overview of the metabolic pathways of AFB1 is presented in Figure 1 [7].

Seventy percent of the global human immunodeficiency virus (HIV) patient population is located in the same region where the AFB1 prevalence is high: Sub-Saharan Africa [13]. The standard of care treatment for HIV patients consists of a combination of three or more antiretroviral (ARV) drugs. Since HIV drugs are taken daily by HIV patients, and food is contaminated with AFB1, there is a possibility for food contaminant–drug interactions. However, to date, food contaminant–drug interactions are understudied, notwithstanding the realistic risk of interactions. Efavirenz (EFV) is a non-nucleoside reverse transcriptase inhibitor used in HIV therapy. EFV undergoes hepatic metabolism mediated by CYP1A2, CYP2A6, CYP2B6 and CYP3A4. Additionally, EFV acts as a CYP2B6, CYP2C19, and CYP3A4 inducer. The induction of CYP3A4 by EFV may potentially lead to interactions with AFB1 since it is partially metabolized by CYP3A4 [14]. Since human in vivo trials with AFB1 are not allowed due to ethical constraints, a physiologically based pharmacokinetic (PBPK) model was built for AFB1 using SimCYP (SimCYP Ltd., a Certara company, Sheffield, UK, version 21) [15]. Drugs will rather have an impact on mycotoxin metabolism than vice versa, considering the magnitude of the ‘dose’ differences [15]. To obtain insight into the possible impact of EFV on the pharmacokinetics (PK) of AFB1, PBPK simulations were performed where AFB1 and EFV were co-administered, and also in vitro experiments were carried out where AFB1 and EFV were co-incubated. The selected in vitro systems were human liver microsomes (HLM) since the focus of this study was on interactions with CYP450 enzymes, for which microsomes are a homogeneous enzyme source [16,17].

## 2. Results

### 2.1. SimCYP Simulations

The results of the different simulations, as depicted in Figure 3 (see Section 5), in the Black South African (BSA) and North European Caucasian (NEC) populations, respectively, are presented in Table 1, Table 2, Table 3 and Table 4.

Based on the ratios presented in the last column of Table 1, it is observed that a single dose exposure to both AFB1 and EFV does not affect the PK of AFB1. This conclusion is supported by the similarity observed in C_max,_ T_max_, AUC, and CL. The ratios between exposure to AFB1 and EFV and AFB1 alone were all between 0.94 and 1.06. 

A daily single exposure to both AFB1 and EFV, as presented in Table 2, clearly indicated a fold change in C_max_, T_max_, AUC, and CL. The ratios of co-administration (AFB1 + EFV) versus AFB1 alone were 0.65, 0.81, 0.39, and 2.58, respectively, in the BSA population and 0.72, 0.87, 0.49, and 2.06, respectively, in the NEC population. In both populations, a decrease in C_max_, T_max,_ and AUC is predicted when AFB1 is co-administered with EFV. The CL increases at least 2.06-fold in both populations. Based on the predictions, the impact of EFV on the PK of AFB1 is more pronounced in the BSA population compared to the NEC population (i.e., 10% lower C_max_, 7% lower T_max_, 21% lower AUC, and 25% higher CL).

Treatment administration, as in Table 3, shows that EFV, when the steady state is achieved, has an impact on the PK of a single AFB1 dose in both populations. The highest fold-change is predicted in the BSA population with a C_max_, T_max_, AUC, and CL ratio of 0.76, 0.83, 0.50, and 2.01, respectively. In the NEC population, these ratios were 0.87, 0.90, 0.62, and 1.61, respectively.

The simulated PK results in the BSA and NEC population, where 30 ng of AFB1 with a combination of different HIV antivirals was co-administered on a daily basis, provided in Table 4, show the highest impact on the PK of AFB1 in the BSA population. Ratios of 0.67–0.74, 0.81–0.88, 0.41–0.51, and 2.46–1.97 were observed for C_max_, T_max_, AUC, and CL, respectively, in the two populations.

The simulated fractions metabolized (f_m_) by CYP1A2, CYP3A4, and renally excreted in the absence and presence of EFV are displayed in Table 5 after both single and multiple dosing.

The difference in fractional contribution to the elimination of AFB1 in the absence and presence of EFV in the BSA population and the NEC population after multiple dosing is graphically represented in Figure 2.

### 2.2. Co-Incubation of Efavirenz and Aflatoxin B1 in Pooled Human Liver Microsomes

Incubations in HLM with midazolam (MDZ) in the presence and absence of EFV, 0.15 µM and 5 µM, showed that CYP3A4 was not induced at 0.15 µM since metabolite formation was similar to samples not exposed to EFV (1.12-fold difference). When exposed to 5 µM of EFV, a higher formation of 1′-OH-MDZ (2.9-fold difference) was observed, implying induction of CYP3A4. Samples exposed to MDZ in the presence of ketoconazole showed a 94% decrease in 1′-OH-MDZ formation, verifying that CYP3A4 is responsible for the metabolism of MDZ into 1′-OH-MDZ. Henceforth, the selected concentration of EFV in the co-incubation assays involving AFB1 was 5 µM. The four control samples did not show an impact on the substrate concentration nor on its solubility. The amounts of AFL, AFM1, and AFQ1 formed in the sample are displayed in Table 6.

The results reveal that simultaneous exposure to both AFB1 and EFV (5 µM) leads to a similar formation of AFL and a small reduction in AFM1 and AFQ1 formation. However, the decrease in AFB1 showed that co-incubation with EFV led to an increased biotransformation of AFB1. The sum of AFL, AFM1, and AFQ1 accounted for 16% of the observed decrease of AFB1 in samples only exposed to AFB1 and for 11% in samples exposed to both AFB1 and EFV.

## 3. Discussion

### 3.1. SimCYP Simulations

#### 3.1.1. Research Premise 1: Does Single Exposure to EFV Impact the PK of AFB1 When Acutely Exposed?

The results from the single-dose exposure to AFB1 and EFV in both BSA and NEC populations clearly show that EFV has no impact on the PK of AFB1. This can be explained since EFV has not attained steady state induction conditions following a single exposure to 600 mg EFV and is consequently not inducing CYP3A4. A study by Bettonte et al., 2023 observed that at least 14 days of exposure are necessary to reach the maximal induction effect of CYP3A4 [18]. Therefore, it can be stated that single-dose exposure to 600 mg of EFV will not impact the PK of AFB1 when acutely exposed.

#### 3.1.2. Research Premise 2: Does Daily Exposure to EFV Impact the PK of AFB1 When Daily Exposed for 30 Days?

After daily exposure to AFB1 in combination with a daily administration of 600 mg EFV for 30 days, a clear impact on the PK of AFB1 was observed in the PBPK model predictions. In the BSA cohort, after co-exposure to both AFB1 and EFV for 30 days, there was a 35% decrease in C_max_, a 19% reduction in the time required to reach C_max_, a substantial 61% decrease in the AUC, and a notable 158% increase in CL. In the NEC cohort, a similar outcome was observed, but with a 28%, 13%, and 51% decrease in C_max_, T_max_, and AUC, respectively, as well as a doubling of the CL. The higher CL indicates a higher metabolic rate and/or an increased excretion. However, since EFV is a CYP3A4 inducer, it is expected that the interaction occurs via the metabolism of AFB1. The impact of EFV on the PK of AFB1 was more pronounced in the BSA population, which can be attributed to differences in CYP450 abundancies in the BSA and NEC populations. Four CYP450 enzymes are involved in the metabolism of EFV, namely CYP1A2, CYP2A6, CYP2B6, and CYP3A4. The abundances of the mentioned CYP450 enzymes are 52, 6.9, 20, and 137 pmol/mg protein in the BSA population and 52, 21.6, 29.3, and 137 pmol/mg in the NEC population, respectively. Additionally, CYP3A4 is also a crucial enzyme in the metabolism of AFB1. With CYP2A6 and CYP2B6 being (slightly) less abundant in the BSA population, it is anticipated that EFV will have a slightly longer half-life in this population, with a higher level of induction of both CYP2B6 and CYP3A4, resulting in a higher metabolic rate of AFB1, and a lower C_max_, lower AUC, and higher CL. Using SimCYP, a half-life of 49.06 h was predicted in the NEC population, and a half-life of 64.58 h was predicted in the BSA population after an oral dose of 600 mg EFV, which is in line with CYP2A6 and CYP2B6 being less abundant in the BSA population [5].

#### 3.1.3. Research Premise 3: Does Daily Exposure to EFV Impact the PK of AFB1 When Acutely Exposed?

Simulations with a single, acute exposure to AFB1 and a daily, multiple-dose administration of EFV for 30 days showed that chronic administration of EFV leads to a higher CL of AFB1 in both populations, up to a 101% increase in the BSA population and a shift from CYP1A2 as the main metabolizing enzyme of AFB1 to CYP3A4. This was to be expected since EFV is classified as a hepatic CYP3A4 inducer [19]. As mentioned previously, a maximal induction of CYP3A4 can be expected after 14 days. [18] Therefore, the simulation results of daily exposure to 600 mg of EFV for 30 days with acute exposure to 30 ng of AFB1 on day 30 align with anticipated results.

#### 3.1.4. Research Premise 4: Does Daily Exposure to EFV/LAM/TFV Impact the PK of AFB1 When Exposed Daily?

A simulation was performed where AFB1 (30 ng), EFV (600 mg), LAM (300 mg), and TFV (300 mg) were administered daily over 30 days. This simulation was performed since EFV is not often used in monotherapy in HIV patients. The combination of the three drugs (EFV/LAM/TFV) is available under the brand name Symfi^®^ (Mylan) and is approved by the Food and Drug Administration (FDA). In South Africa, this therapy is recommended if dolutegravir (DTG)/LAM/TFV cannot be used in case of DTG contra-indication [20]. TFV is not a CYP450 substrate nor an inducer or inhibitor of CYP450 enzymes [21]. Furthermore, LAM is neither a CYP450 substrate nor a CYP450 perpetrator [21]. Hence, when EFV/LAM/TFV (daily intake) is co-administered with AFB1, a comparable impact is predicted on the PK of AFB1 as those predicted for a daily intake with EFV alone.

Next, the fractions metabolized by hepatic CYP3A4, CYP1A2, and renally excreted were compared after both single and multiple dosing and in both the BSA and NEC populations. In all cases, it was observed that the contribution of CYP1A2 lowered from 63.1 to 25.9% in the BSA and from 63.9 to 32.0% in the NEC population, while the fraction metabolized by CYP3A4 increased from 36.8 to 74.0% in the BSA and from 36.0 to 67.9% in the NEC population when co-exposed to 600 mg of EFV daily. These results were again in line with expectations since EFV is a CYP3A4 inducer. 

### 3.2. Co-Incubation of Efavirenz and Aflatoxin B1 in Pooled Human Liver Microsomes

In this study, a food contaminant–drug interaction study was performed. No official guidelines are available to perform this interaction. Therefore, we opted to follow the guidelines provided by the International Council on Harmonization (ICH), European Medicines Agency (EMA), and FDA on drug–drug interaction studies (DDI). Considering the focus on metabolism interactions with CYP450 enzymes, HLM was selected as an in vitro system for the performance of this study [16,17,22]. A more thorough understanding of the PK of AFB1 could be obtained if monitoring encompassed all metabolites. To date, AFL, AFM1, and AFQ1 are commercially available for use; however, analytical reference standards for AFBO are too unstable to be used in co-incubation experiments due to their high reactivity. Consequently, the in vitro analysis concerning AFBO formation and downstream products was impossible to conduct. In humans, AFB1 undergoes conversion mediated by CYP1A2 and CYP3A4, resulting in the formation of endo-AFBO and exo-AFBO, among others. Subsequently, AFBO will lead to a variety of metabolites (Figure 1) [4]. AFBO will form DNA adducts leading to AFB-N^7^-guanine (AFB-N^7^-gua) and aflatoxin B1-formamidopyrimidine (AFB1-FAPyr). Additionally, AFBO is metabolized into aflatoxin B-S-glutathione (AFB-GSH) and aflatoxin-mercapturic acid (AFB-NAC). Further metabolism of AFBO yields AFB-diol and AFB-dialdehyde, with the latter reacting with serum albumin to form AFB-lysine. Furthermore, AFB-monoalcohols and AFB-dialcohol are also formed [4,5,23].

When analyzing the results from the in vitro experiments with MDZ, exposure to 0.15 µM of EFV did not activate CYP3A-mediated metabolism [24]. The concentration of 0.15 µM is similar to the free plasma concentration of EFV after single-dose exposure to 600 mg of EFV. This aligns with the outcomes of the PBPK simulations conducted for a single oral dose of 600 mg of EFV, where no difference in the PK of AFB1 was predicted. The formation of 1′-OH-MDZ is facilitated by CYP3A4. This was experimentally validated using ketoconazole, a CYP3A4 inhibitor, significantly reducing the formation of 1′-OH-MDZ. Exposure to 5 µM of EFV induced CYP3A4 activity, leading to an increased 1′-OH-MDZ formation and thus impacting metabolite formation. Therefore, co-incubation experiments with both EFV and AFB1 were performed with 5 µM of EFV. The formation of AFL accounted for 2.25% and 1.86% of the metabolite formation in incubations with AFB1 alone and with AFB1 and EFV combined, respectively. For AFM1, the percentages were 5.47% and 3.52%, while for AFQ1 formation, they were 8.04% and 5.68% of the metabolite formation in incubations with AFB1 alone and with AFB1 and EFV combined, respectively. For both single and co-incubations, it can be concluded that AFL, AFM1, and AFQ1 only accounted for 11–16% of the metabolite formation. Although these pathways are referred to as the major metabolic pathways in the human biotransformation of AFB1 [25,26], it seems that the majority of metabolites are not captured. Co-incubation experiments with AFB1 and EFV showed modest changes in AFL, AFM1, and AFQ1 formation, indicating these pathways may not fully represent AFB1 metabolism. The authors acknowledge inherent limitations with the setup of the present study since not all metabolites were monitored (AFP1, AFH1, AFBO, and other downstream metabolites), while these metabolites will most probably account for another major part of the metabolite formation. Recent research in HLM and primary human hepatocytes indicated AFL, AFQ1, and AFM1 a.o. as the major metabolic pathways in the human biotransformation of AFB1 25,26]. In the work of Slobodchikova et al., 2019 metabolites formed in vitro using HLM were monitored using liquid chromatography high-resolution mass spectrometry (LC-HRMS) [26]. A variety of metabolites were observed but did not account for the decrease in AFB1 (40% of AFB1 decreased) observed in the incubated samples. The conditions were similar to the conditions of the experiments performed in this research, and the decrease in AFB1 was also around 40%. It can be expected that the number of moles of AFB1 that disappear after incubation will be recovered as the sum of the number of moles originating from metabolites. The recovery of AFB1 decrease has not yet been retrieved, but insolubility issues of AFB1, degradation of AFB1 due to sample treatment, or poor recovery of the analytical method were ruled out in this research by using an in-house validated method and by incorporating control samples. Therefore, the reduction observed in AFB1 levels can reasonably be attributed to biotransformation. Since the decrease in AFB1 is similar to the experiments of Slobodchikova, it is likely to rule out experimental setup factors for the achievement of these results [26]. Also, EFV activated the CYP3A-mediated midazolam 1′-hydroxylation in vitro, and based on the decrease in AFB1, it also had an activating effect on the CYP3A-mediated metabolism of AFB1 [24]. Since no increase was observed in the formation of AFQ1, it can be hypothesized that more AFBO was formed due to CYP3A4 induction or that AFQ1 underwent further conversion to AFH1. Additionally, the potential involvement of CYP3A4 in alternative pathways or the contribution of other metabolic routes in the biotransformation of AFB1 could be implicated, considering the capacity of EFV to also induce CYP2B6 and CYP2C19 [27]. Furthermore, a similar downstream pathway, as for AFBO, can be theoretically expected with the formation of AFM1 epoxides [28]. The binding of AFBO with DNA and albumin is also expected in hepatocytes but not in HLM because of the negligible presence of DNA and albumin. Nonetheless, AFBO might react with proteins present in the sample other than CYP450 enzymes, such as structural and membrane proteins of microsomes [29]. The formation of AFB1 and AFM1 radicals that react with proteins in the sample is also hypothesized [30]. Currently, research has been performed on the metabolism of AFB1, both in vitro and in vivo, but more research is a prerequisite to fully assign the observed AFB1 decrease to metabolites [25,26,31]. Understanding these interactions is crucial, especially for HIV patients consuming AFB1-contaminated food, as EFV could influence AFB1 immunotoxicity. 

Concerning the PBPK modeling, a decrease in CYP1A2 involvement was predicted at 37.2%, i.e., from 63.1% when exposed to AFB1 alone to 25.9% when co-exposed to both AFB1 and EFV in the BSA population. In the NEC population, the predicted involvement of CYP1A2 dropped from 63.9% to 32% after co-administration of EFV. It must be taken into account that CYP1A2 has a formation ratio of AFBO:AFM1 of approximately 2.5:1 [32]. CYP3A4 has a formation ratio of AFQ1:AFBO of approximately 10:1 [32]. EFV is a CYP3A4 inducer, potentially leading to a lower formation of AFM1 and a higher formation of AFQ1, the latter being a detoxification metabolite since it is 18x less toxic and 83x less mutagenic compared to AFB1 [3,5,6,7]. The results of the PBPK modeling were only partially confirmed by the in vitro experiments. However, there might also be a methodological contribution to the observed outcomes. The PBPK simulations did not predict an increase in CYP3A4 metabolism in the case of a single dose of AFB1 and EFV, which was also the case in the in vitro experiments. In the single dose simulations, an oral dose of 30 ng AFB1 and 600 mg EFV was used, whereas in the in vitro simulations, HLM were exposed to 312 ng/mL AFB1 and 50 ng/mL EFV. In order to confirm the PBPK predictions, an in vitro experiment should be performed with chronic exposure to both components, as simulated. However, this is not possible when using HLM. In this research, a higher concentration of EFV was also used to mimic the effect of chronic exposure. While we cannot corroborate chronic exposure in vitro, the simulation results remain interesting. If the formation ratios from the literature [16,19] are applied to the fractions metabolized by CYP3A4 or CYP1A2 from the simulations, exposure to AFB1 alone in the BSA population leads to 48.4% carcinogenic AFBO formation, 33.6% AFQ1 formation and 18% AFM1 formation (considering total metabolite formation amount to 100%). When the same subjects are chronically exposed to AFB1, and EFV is co-administered daily, the AFBO formation lowers to 25.2%, AFQ1 formation increases to 67.3%, and AFM1 formation lowers to 7.4%. Induction of a CYP450 enzyme in a pathway potentially leads to an increased metabolic activity within that specific pathway. However, the impact on metabolites from other pathways is not consistent and may vary. In certain cases, the induction of a specific pathway may lead to competition for the enzyme, potentially resulting in reduced metabolite formation in other pathways. The magnitude of this effect is contingent upon the specific enzymes involved, the substrate characteristics, and the overall complexity of the metabolic network. Additional research is warranted to elucidate the impact of CYP3A4 induction on the other metabolic pathways, e.g., AFBO formation in the metabolism of AFB1 in humans, since in vitro experiments in HLM did not show an increase in AFQ1 formation. In contrast, it showed a decrease in both AFM1 and AFQ1 formation and a stable formation of AFL. 

Noteworthy, AFBO is the general name for both AFB1-exo-8,9-epoxide and AFB1-endo-8,9-epoxide, with the exo-isomer being responsible for genotoxicity [32,33,34,35]. CYP3A4 merely forms AFQ1 and, to a lesser extent, AFBO-exo-isomers, while CYP1A2 is responsible for the formation of AFM1 and AFBO-exo- and endo-isomers [8,33,36,37]. Remarkably, HIV patients have a compromised immune system, and exposure to immunotoxic AFB1 further worsens the capacity of the immune system. The intake of EFV not only plays a crucial role in treating HIV patients, but it might also be relevant for the immunotoxicity caused by AFB1, originating from contaminated food consumption. Further in-depth research needs to be performed to fully disentangle which metabolites are formed in both the breakdown of AFB1 in humans and the altered metabolism of AFB1 when co-incubated with EFV.

## 4. Conclusions

Based on the simulations and the complementary in vitro experiments, chronic AFB1 exposure and daily EFV intake will lead to the formation of less AFM1, a toxic metabolite of AFB1. It is suggested that AFQ1 will be formed in higher quantities via activation of CYP3A-mediated metabolism and that AFBO will also be produced to a lesser extent when taking the formation ratios via CYP1A2 and CYP3A4 from the literature into account. The latter could not be confirmed in the current in vitro experiments; the formation of AFQ1 slightly decreased, and due to instability and commercial unavailability of reference materials, AFBO could not be monitored. Therefore, no conclusions can be drawn from this study on the impact of EFV on AFBO formation. Consequently, it is postulated that individuals with HIV who are undergoing daily EFV therapy and experience (chronic) exposure to AFB1 would exhibit an impact on the PK of AFB1. Further research is needed to elucidate the impacted pathways. Due to ethical constraints in performing human in vivo experiments with AFB1, greater understanding, and elucidation can be achieved through conducting in vitro experiments using primary human cultured hepatocytes. Additional animal trials may provide further insights into in vivo occurrences. It can be concluded that in vitro experiments in combination with PBPK modeling are a useful tool to simulate possible interactions that take place in vivo when human in vivo trials are not possible.

## 5. Materials and Methods

### 5.1. SimCYP Simulations

Simulations were performed using the SimCYP (v21) software. First, a trial simulation was performed in a BSA population of 5000 subjects. The gender distribution was equally balanced, with an age range of 20–50 years. Subjects were exposed to a single dose of 30 ng AFB1 [15], which equals one-twentieth of the US maximum limit of 20 µg/kg AFs in a 30 g peanut butter sandwich [38], combined with a single dose of 600 mg EFV (labeled dose to treat HIV) [39]. Next, the same simulation was performed in an NEC population since commercially available human liver microsomes, employed in in vitro experiments as detailed in Section 5.2, are from Caucasian subjects (Figure 3i).

Another simulation was performed with a similar trial setup but with repeated administration, i.e., exposure to 30 ng AFB1 (once daily) and to 600 mg EFV (once daily) over 30 days in both the BSA and the NEC populations (Figure 3ii).

Next, to verify whether EFV therapy has an influence on acute, single AFB1 exposure, a simulation was performed in both the BSA and the NEC populations where the subjects were orally dosed with 600 mg/day EFV over 30 days, and where an acute exposure to 30 ng AFB1 occurred on day 30 (Figure 3iii).

Lastly, since combination treatment is commonly employed in HIV therapy, a simulation was performed with AFB1 (30 ng), EFV (600 mg), lamivudine (LAM) (300 mg), and tenofovir (TFV) (300 mg). The latter are both nucleoside reverse transcriptase inhibitors (NRTI), often co-administered with EFV. The trial setup was the same, with daily co-exposure to AFB1, EFV, LAM, and TFV over 30 days in both the BSA and the NEC populations (Figure 3iv). For each simulation (i–iv), the following PK parameters were determined: maximum plasma concentration (C_max_), the time at which C_max_ is achieved (T_max_), the area under the curve (AUC) during the dosing interval of the mean plasma concentration–time profile, and the clearance (CL).

**Figure 3 toxins-16-00259-f003:**
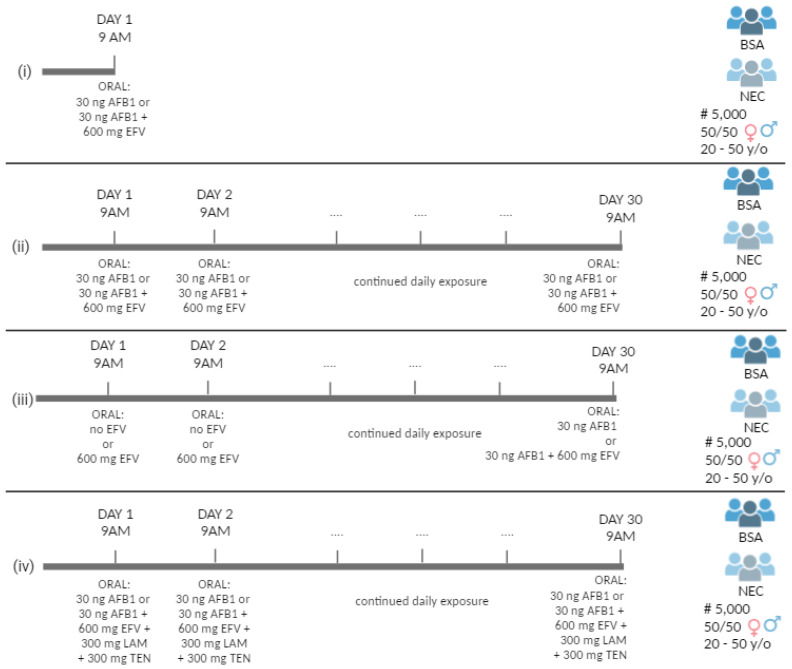
Overview of the different dosing schemes used in physiologically based pharmacokinetic (PBPK) modelling in a Black South African (BSA) population and a North European Caucasian (NEC) population exposed to four different scenarios being (**i**) a single dose of 30 ng aflatoxin B1 (AFB1) alone or co-administration of a single dose of 30 ng AFB1 and 600 mg efavirenz (EFV); (**ii**) 30-day dosing of 30 ng AFB1 alone or co-administration of 30 ng AFB1 and 600 mg EFV during 30 days; (**iii**) no EFV and a single dose of 30 ng AFB1 alone on day 30 or daily administration of 600 mg of EFV for 30 days and a single dose of 30 ng AFB1 on day 30; and (**iv**) 30-day dosing of 30 ng AFB1 alone or co-administration of 30 ng AFB1 and 600 mg of EFV, 300 mg of lamivudine (LAM) and 300 mg of tenofovir (TFV) on a daily basis over 30 consecutive days. The number of subjects in the simulated population is shown with #; y/o is an abbreviation for years old.

### 5.2. Co-Incubation of Efavirenz and Aflatoxin B1 in Pooled Human Liver Microsomes

In vitro, co-incubation experiments were performed with AFB1, midazolam (MDZ), i.e., a CYP3A4 probe substrate, and EFV to analyze the effect of EFV on the metabolism of AFB1 and to verify whether induction of CYP3A4 was achieved. AFB1, AFL, and zearalanone (ZAN), used as internal standard (IS) for AFB1, were purchased from Fermentek (Jerusalem, Israel). AFM1 was retrieved from Romer Labs (Getzersdorf, Austria), and AFQ1 was obtained from Toronto Research Chemicals (Toronto, ON, Canada). EFV, ammonium acetate, and formic acid (FA) were acquired from Merck (Darmstadt, Germany). MDZ and 1′-hydroxy-MDZ (1′-OH-MDZ) were retrieved from Roche (Mannheim, Germany). Chlorpropamide (CHL), used as IS for MDZ, and ketoconazole, a CYP3A4 inhibitor, were acquired from Sigma Aldrich (St. Louis, MO, USA). Human liver microsomes (HLM) (Corning^®^ UltraPool™ HLM 150, Mixed Gender, 0.5 mL), for experimental use, were purchased from Corning (Woburn, MA, USA). Information on the preparation of the HLM was detailed in the product description sheet. The HLM were stored at −80 °C upon usage. Water was obtained from an ultrapure water system (Sartorius, Goettingen, Germany). Nicotinamide adenine dinucleotide phosphate tetrasodium (NADPH.4Na) was retrieved from Gentaur (Kampenhout, Belgium) and stored at −20 °C. Dipotassium hydrogen phosphate, potassium dihydrogen phosphate, and potassium chloride (KCl) were purchased from VWR (Oud-Heverlee, Belgium). Acetonitrile (ACN), LC-MS grade methanol (MeOH), and glacial acetic acid were acquired from Biosolve B.V. (Valkenswaard, The Netherlands). All chemicals and reagents were of analytical grade. SimCYP (Certara, version 21) was used for the prediction of co-incubations in humans.

Stock solutions of EFV (20 mg/mL), AFB1 (1 mg/mL), AFM1 (0.1 mg/mL), AFQ1 (1.096 mg/mL), AFL (0.1 mg/mL), MDZ (1 mg/mL), 1′-OH-MDZ (100 µg/mL), CHL (1 mg/mL), ketoconazole (1 mg/mL), and ZAN (1 mg/mL) were prepared in MeOH and stored at −20 °C (AFB1 at 4 °C). Work solutions of AFB1, AFL, AFM1, and AFQ1 were made in ultrapure water at a concentration of 5 µM for AFB1 and at a concentration of 0.025–0.5 µM for AFL, AFM1, and AFQ1; the selected concentrations were based on the limit of quantification (LOQ) of the developed LC-MS/MS method. Work solutions of MDZ and 1′-OH-MDZ were made in ultrapure water at concentrations from 0.5 to 10 µM. Work solutions of EFV were made in MeOH, in view of its low aqueous solubility, at a concentration of 7.9 µM and 250 µM. The concentration in the sample exposed to 7.9 µM EFV is 50 ng/mL, similar to the free plasma concentration of EFV after 600 mg oral dosing for single-dose exposure experiments, and the other samples were exposed to 5 µM EFV, inducing CYP3A4 [24,40,41]. Phosphate buffer, pH 7.4, with a concentration of 0.2 M, was prepared and frozen at −20 °C in 60 mL tubes. NADPH.4Na (5 mM) was freshly made every experimental day at a concentration of 5 mM in phosphate buffer (pH 7.4). A 1.15% KCl solution was made in ultrapure water and stored at 4 °C. HLM (20 mg/mL in 250 mM sucrose) was diluted in 1.15% KCl to achieve a final protein concentration of 0.5 mg/mL in the samples. The stop reagent was made with 300 µL FA, 5500 µL ACN, and ultrapure water (diluted to 10 mL) and contained the internal standard (IS) (sample end concentration of 0.12 µg/mL ZAN for AFB1 and 0.02 µg/mL CHL for MDZ). 

Samples were prepared based on the protocol detailed in Lootens et al., 2022 [31]. In brief, centrifugal Eppendorf cups were filled with 50 µL substrate solution (i.e., 5 µM for AFB1 or 5 µM for MDZ) and with 50 µL of 1.15% KCl or 5 µL of co-substrate EFV (7.9 µM or 250 µM) and 45 µL of 1.15% KCl solution, in case of co-incubation. Additionally, 50 µL of 0.2 M phosphate buffer was added. To confirm that the metabolism of MDZ into 1′-OH-MDZ can be assigned to CYP3A4 metabolism in HLM, three samples were subjected to incubation in the presence of 5 µL ketoconazole (50 µM), a potent CYP3A4 inhibitor. To rule out certain factors, such as the impact of used solutions on the substrate concentration or solubility issues, four control samples were analyzed. The control samples consisted of the same solutions as the tested samples with an alteration, i.e., control 1 without substrate solution, control 2 without NADPH, control 3 with heated microsomes (20 min at 37 °C), and control 4 without HLM. Next, 50 µL of a freshly prepared NADPH.4Na solution was added. After an incubation of 3 min, 50 µL of diluted HLM (0.25 mg/mL for MDZ and 2.5 mg/mL for AFB1) were added and placed back on the thermoshaker TS-100 (Biosan, Geraardsbergen, Belgium) with a rotation speed of 300 rpm, at a temperature of 37 °C. After the indicated time (i.e., 5 min for MDZ, 20 min for AFB1), 25 µL of an ice-cold stop reagent with IS was added. Samples were centrifuged at 16,000× *g* for 20 min at 4 °C and transferred into vials; all samples were made in triplicate. An aliquot of 5 µL was injected into the ultrahigh performance liquid chromatography (UPLC)—XEVO TQ-S tandem quadrupole mass spectrometer (MS/MS) equipment (Waters, Milford, MA, USA) using an in-house developed and validated method (see Section 5.3).

### 5.3. LC-MS/MS

A Waters Acquity class I UPLC system coupled to a XEVO TQ-S MS/MS from Waters (Milford, MA, USA) was used for the detection and quantification of AFB1, AFL, AFM1, AFQ1, ZAN, MDZ, 1′-OH-MDZ, and CHL. A Charged Surface Hybrid (CSH) C_18_ column (1.7 µM 2.1 × 100 mm) with a Guard column was used for chromatographic separation, and the column temperature was set at 30 °C; the sample temperature was set at 10 °C. The mobile phase was used at a flow rate of 0.250 mL/min following a gradient program (Appendix A). The total duration for a single run was 12 min. The first minute of the run was sent to the waste to eliminate the phosphate buffer. The MS was operated in the electrospray positive mode (ESI+) with multiple reaction monitoring (MRM). The MRM parameters of AFB1, AFL, AFM1, AFQ1, ZAN, MDZ, 1′-OH-MDZ, and CHL, respectively, are depicted in Appendix A. The source temperature and desolvation temperature were set at 130 °C and 200 °C. The cone gas flow was set at 150 L/h, and the desolvation gas flow rate at 550 L/h. Data were processed using the Masslynx and Targetlynx software from Waters (Manchester, UK).

## Figures and Tables

**Figure 1 toxins-16-00259-f001:**
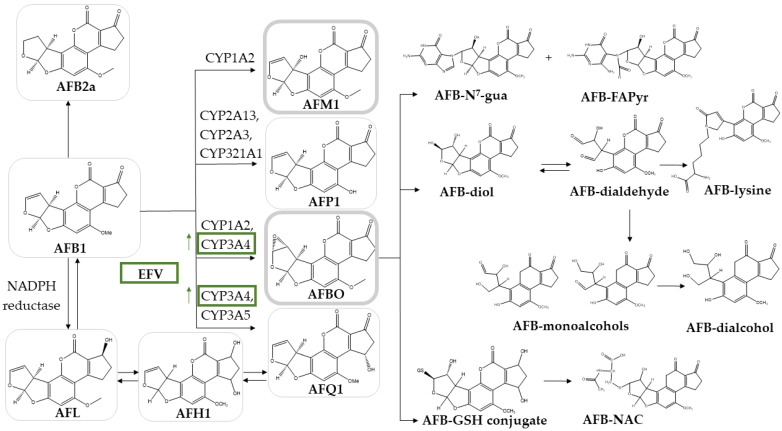
Human metabolic pathways of aflatoxin B1 (AFB1). Aflatoxin M1 (AFM1) is formed by CYP1A2, aflatoxin P1 (AFP1) by CYP2A13, CYP2A3, and CYP321A1. Aflatoxin-8,9-endo/exo-epoxides (AFBO) are formed by both CYP1A2 and CYP3A4. More downstream metabolites of AFBO are depicted without a frame (aflatoxin B-N^7^-guanine (AFB-N^7^-gua), aflatoxin B-formamidopyridine (AFB-FAPyr), aflatoxin B-diol (AFB-diol), aflatoxin B-dialdehyde (AFB-dialdehyde), aflatoxin B-S-glutathion (AFB-GSH), aflatoxin B-mercapturic acid (AFB-NAC), aflatoxin B1-lysine (AFB-lysine), aflatoxin B-monoalcohols (AFB-monoalcohols), and aflatoxin B dialcohol (AFB-dialcohol)). Aflatoxin B2a (AFB2a) is formed by the involvement of CYP450 enzymes, not further specified. Aflatoxin Q1 (AFQ1) is formed by CYP3A4 and CYP3A5. Aflatoxicol (AFL) is produced by nicotinamide-adenine-dinucleotide phosphate reductase (NADPH reductase). Both AFL and AFQ1 can form aflatoxicol H1 (AFH1). Efavirenz (EFV), a non-nucleoside reverse transcriptase inhibitor drug and CYP3A4 inducer, is also presented, indicating its inducing effect on CYP3A4, expected to increase the formation of both AFQ1 and AFBO (green frames) [5]. The toxic metabolites of AFB1 are presented in thick grey frames (AFM1 and AFBO).

**Figure 2 toxins-16-00259-f002:**
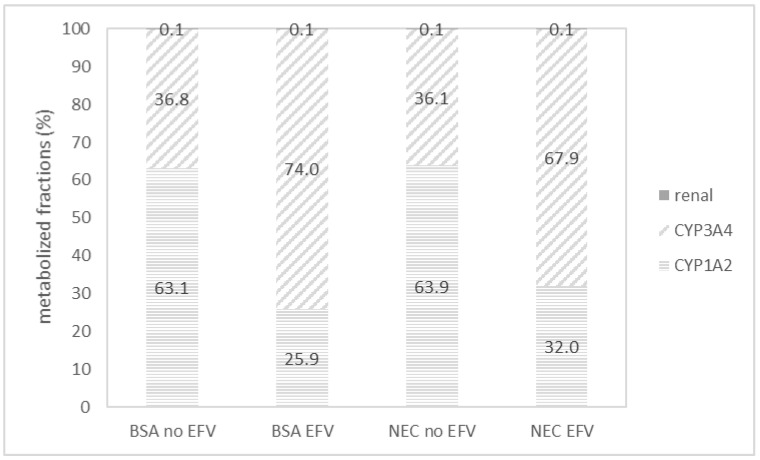
Graphical representation of the varying fractions of aflatoxin B1 (AFB1) metabolized by hepatic CYP1A2 (indicated by horizontal lines) and CYP3A4 enzymes (indicated by diagonal lines), along with renal elimination (depicted in grey), after multiple dosing. The Black South African population is denoted as BSA, while the North European Caucasian population is denoted as NEC in the *x*-axis. The absence of efavirenz (EFV) is indicated as ‘no EFV’, while its presence is indicated as ‘EFV’.

**Table 1 toxins-16-00259-t001:** Predicted geometric mean values of pharmacokinetic parameters of AFB1 (+90% confidence interval (CI)) obtained using physiologically based pharmacokinetic modeling in a Black South African (BSA) population (in orange) and a North European Caucasian (NEC) population (in blue) exposed to (i) 30 ng aflatoxin B1 (AFB1) alone and co-exposure to both 30 ng of AFB1 and 600 mg of efavirenz (EFV) during 24 h.

	*(i)* *A Single Dose of 30 ng Aflatoxin B1 (AFB1) Alone or Co-Administration of a Single Dose of 30 ng AFB1 and 600 mg Efavirenz (EFV)*				
	AFB1 (30 ng) Alone	[90% CI]	AFB1 (30 ng) + EFV (600 mg)	[90% CI]	Ratio $\frac{AFB1+EFV}{AFB1\;alone}$
	BSA				
C_max_ (pg/mL)	0.706	[0.700–0.712]	0.703	[0.697–0.710]	1
T_max_ (h)	1.55	[1.54–1.57]	1.54	[1.53–1.55]	0.99
AUC (pg/mL.h)	4.68	[4.59–4.76]	4.42	[4.34–4.50]	0.94
CL (L/h)	6.42	[6.30–6.53]	6.79	[6.67–6.91]	1.06
	NEC				
C_max_ (pg/mL)	0.709	[0.704–0.714]	0.706	[0.701–0.711]	1
T_max_ (h)	1.59	[1.58–1.60]	1.58	[1.57–1.59]	0.99
AUC (pg/mL.h)	5.66	[5.58–5.74]	5.37	[5.30–5.45]	0.95
CL (L/h)	5.30	[5.22–5.38]	5.58	[5.50–5.66]	1.05

C_max_—maximum plasma concentration; T_max_—time when C_max_ is achieved; AUC—area under the curve during the interval of the mean concentration–time profile; CL—clearance.

**Table 2 toxins-16-00259-t002:** Predicted geometric mean values of pharmacokinetic parameters of AFB1 (+90% confidence interval (CI)) obtained using physiologically based pharmacokinetic modelling in a Black South African (BSA) population (in orange) and a North European Caucasian (NEC) population (in blue) exposed to (ii) 30-day dosing of 30 ng AFB1 alone and co-exposure to 30 ng of AFB1 and 600 mg EFV during 30 days.

	*(ii)* *30-Day Dosing of 30 ng AFB1 Alone or Co-Administration of 30 ng AFB1 and 600 mg EFV during 30 days*				
	AFB1 (30 ng) Alone	[90% CI]	AFB1 (30 ng) + EFV (600 mg)	[90% CI]	Ratio $\frac{AFB1+EFV}{AFB1\;alone}$
	BSA				
C_max_ (pg/mL)	0.956	[0.882–1.03]	0.624	[0.578–0.673]	0.65
T_max_ (h)	1.72	[1.65–1.81]	1.39	[1.33–1.46]	0.81
AUC (pg/mL.h)	8.34	[7.15–9.73]	3.23	[2.79–3.74]	0.39
CL (L/h)	3.60	[3.08–4.20]	9.29	[8.02–10.8]	2.58
	NEC				
C_max_ (pg/mL)	0.955	[0.891–1.02]	0.691	[0.651–0.734]	0.72
T_max_ (h)	1.63	[1.55–1.71]	1.42	[1.36–1.49]	0.87
AUC (pg/mL.h)	9.71	[8.54–11.0]	4.72	[4.17–5.33]	0.49
CL (L/h)	3.09	[2.72–3.51]	6.36	[5.63–7.19]	2.06

C_max_—maximum plasma concentration; T_max_—time when C_max_ is achieved; AUC—area under the curve during the interval of the mean concentration–time profile; CL—clearance.

**Table 3 toxins-16-00259-t003:** Predicted geometric mean values of pharmacokinetic parameters of AFB1 (+90% confidence interval (CI)) obtained using physiologically based pharmacokinetic modelling in a Black South African (BSA) population (in orange) and a North European Caucasian (NEC) population (in blue) exposed to (iii) a single AFB1 exposure alone on day 30 and daily administration of 600 mg of EFV for 30 days and single exposure to 30 ng AFB1 on the day.

	*(iii)* *No EFV and a Single Dose of 30 ng AFB1 Alone on Day 30 or Daily Administration of 600 mg of EFV for 30 Days and a Single Dose of 30 ng AFB1 on Day 30*				
	AFB1 (30 ng) Alone	[90% CI]	AFB1 (30 ng) + EFV (600 mg)	[90% CI]	Ratio $\frac{AFB1+EFV}{AFB1\;alone}$
	BSA				
C_max_ (pg/mL)	0.756	[0.708–0.808]	0.573	[0.533–0.616]	0.76
T_max_ (h)	1.63	[1.55–1.72]	1.35	[1.29–1.42]	0.83
AUC (pg/mL.h)	5.45	[4.76–6.25]	2.72	[2.38–3.11]	0.50
CL (L/h)	5.50	[4.80–6.30]	11.03	[9.63–12.6]	2.01
	NEC				
C_max_ (pg/mL)	0.911	[0.879–0.946]	0.797	[0.766–0.828]	0.87
T_max_ (h)	0.94	[0.89–0.99]	0.85	[0.81–0.89]	0.90
AUC (pg/mL.h)	5.82	[5.26–6.44]	3.62	[3.28–4.00]	0.62
CL (L/h)	5.15	[4.66–5.71]	8.28	[7.50–9.14]	1.61

C_max_—maximum plasma concentration; T_max_—time when C_max_ is achieved; AUC—area under the curve during the interval of the mean concentration–time profile; CL—clearance.

**Table 4 toxins-16-00259-t004:** Predicted geometric mean values of pharmacokinetic parameters of AFB1 (+90% confidence interval (CI)) obtained using physiologically based pharmacokinetic modelling in a Black South African (BSA) population (in orange) and a North European Caucasian (NEC) population (in blue) exposed to (iv) 30 ng of AFB1 alone and co-exposure to 30 ng of AFB1 and 600 mg of EFV, 300 mg of lamivudine (LAM) and 300 mg of tenofovir (TFV) on a daily basis over 30 consecutive days.

	*(iv)* *30-Day Dosing of 30 ng AFB1 Alone or Co-Administration of 30 ng AFB1 and 600 mg of EFV, 300 mg of Lamivudine (LAM), and 300 mg of Tenofovir (TFV) on a Daily Basis over 30 Consecutive Days*				
	AFB1 (30 ng) Alone	[90% CI]	AFB1 (30 ng) + EFV/LAM/TFV (600/300/300 mg)	[90% CI]	Ratio $\frac{AFB1+EFV}{AFB1\;alone}$
	BSA				
C_max_ (pg/mL)	0.946	[0.875–1.02]	0.635	[0.586–0.689]	0.67
T_max_ (h)	1.72	[1.64–1.80]	1.40	[1.33–1.47]	0.81
AUC (pg/mL.h)	8.17	[7.02–9.50]	3.33	[2.86–3.88]	0.41
CL (L/h)	3.67	[3.16–4.27]	9.01	[7.74–10.5]	2.46
	NEC				
C_max_ (pg/mL)	0.945	[0.883–1.01]	0.702	[0.659–0.747]	0.74
T_max_ (h)	1.62	[1.55–1.70]	1.43	[1.37–1.50]	0.88
AUC (pg/mL.h)	9.51	[8.38–10.8]	4.83	[4.26–5.47]	0.51
CL (L/h)	3.16	[2.78–3.58]	6.21	[5.49–7.04]	1.97

C_max_—maximum plasma concentration; T_max_—time when C_max_ is achieved; AUC—area under the curve during the interval of the mean concentration–time profile; CL—clearance.

**Table 5 toxins-16-00259-t005:** Predicted metabolism of aflatoxin B1 (AFB1)**:** fractions metabolized (%) by CYP1A2, CYP3A4 and renally excreted after single (SD) and multiple doses (MD) in the absence and presence of efavirenz (EFV) in the Black South African population and the North European Caucasian population. Standard deviations are presented within brackets.

	Black South African				North European Caucasian			
	SD		MD		SD		MD	
	AFB1	AFB1 + EFV	AFB1	AFB1 + EFV	AFB1	AFB1 + EFV	AFB1	AFB1 + EFV
f_m CYP1A2_ (%)	63.7 (15.0)	26.9 (14.4)	63.1 (14.7)	25.9 (14.9)	63.7 (15.2)	31.3 (15.9)	63.9 (15.1)	32.0 (17.4)
f_m CYP3A4_ (%)	36.2 (14.9)	73.1 (14.4)	36.8 (14.6)	74.0 (14.9)	36.2 (15.2)	68.6 (15.9)	36.1 (15.1)	67.9 (17.4)
renal (%)	0.12 (0.03)	0.09 (0.02)	0.12 (0.03)	0.09 (0.02)	0.12 (0.04)	0.10 (0.02)	0.13 (0.04)	0.10 (0.03)

**Table 6 toxins-16-00259-t006:** In vitro formation of aflatoxicol (AFL), aflatoxin M1 (AFM1), and aflatoxin Q1 (AFQ1) in pmol in samples solely exposed to aflatoxin B1 (AFB1) and co-exposed to both AFB1 (1 µM) and efavirenz (EFV) (5 µM). Standard deviations are presented within brackets.

	AFL (pmol)	AFM1 (pmol)	AFQ1 (pmol)	Sum of AFL, AFM1 and AFQ1 (pmol)	Decrease in AFB1 (pmol)	AFL, AFM1 and AFQ1 to Total Metabolite Formation (%)
AFB1 (1 µM)	1.78 (0.267)	4.33 (0.349)	6.37 (0.510)	12.48 (1.03)	79.14 (20.9)	16 (5.0)
AFB1 (1 µM) + EFV (5 µM)	1.84 (0.0875)	3.48 (0.654)	5.61 (0.887)	10.93 (1.61)	98.75 (12.5)	11 (3.0)

## Data Availability

The raw data supporting the conclusions of this article will be made available by the authors upon request.

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
