# Peer review of "Exploring the Impact of Efavirenz on Aflatoxin B1 Metabolism: Insights from a Physiologically Based Pharmacokinetic Model and a Human Liver Microsome Study"

_toxins, 2024, doi:10.3390/toxins16060259_

Round 1
Reviewer 1 Report
Comments and Suggestions for Authors
In this study, Authors aimed to examine the impacts of efavirenz (EFV) on the toxicokinetics of aflatoxin B1 (AFB1) applying physiologically-based pharmacokinetic modeling (PBPK) and in vitro experiments on human liver microsomes. The impacts of drugs and other xenobiotics on the toxicokinetics of mycotoxins is a barely characterized field, which is worthy for investigation. Nevertheless, I have serious problems with the experimental design and the quality of this work. The in vitro model applied seems to be improper, there are some controversial results, and the conclusions are not established. Therefore, I do not suggest the publication of this manuscript. My major critical comments are listed below.
I have major problems with the in vitro study applied. Enzyme induction should be tested on liver cells or slices and not on microsomes. Microsomes are typically used to test enzyme inhibition. In addition, I do not see clearly the experimental design, Authors refer to Lootens et al., 2022; however, in the cited manuscript, inhibitory effects were examined on microsomes and not enzyme induction.
Based on previous studies, EFV is both an inhibitor and an inducer of CYP3A4; however, I do not see that simple experiments of microsomes can give a proper answer regarding the impact of EFV on the toxicokinetics of AFB1. This manuscript seems to be a preliminary study, it is not a complete work and does not provide any evidence (even if Authors state it in the abstract).
Further experiments are highly reasonable, including in vitro studies on primary human cultured hepatocytes and/or animal studies.
Some results are simply controversial. For example, Authors suggest that the CYP3A4-mediated biotransformation of AFB1 increased due to induction; however, the formation of AFQ1 (a metabolite of AFB1 produced by CYP3A4) is lower as a result of EFV co-treatment.
Previous studies demonstrated that CYP3A4 has only limited involvement in the toxic activation of AFB1 if the liver was exposed to nanomolar concentrations of the mycotoxin. CYP3A4-catalyzed and CYP1A2-mediated activations of AFB1 follow Hill kinetics and Michaelis-Menten kinetics, respectively. Therefore, regarding the ‘real world’ AFBO production, CYP1A2 is the dominant enzyme. (https://doi.org/10.1016/j.mrrev.2018.10.002)
Therefore, the importance of CYP3A4-based interactions regarding the toxicokinetics of AFB1 is questionable.
The discussion section only repeats the results and does not provide a proper discussion.
Comments on the Quality of English LanguageOnly minor linguistic corrections are required.
Reviewer 2 Report
Comments and Suggestions for Authors
The manuscript "Exploring the Impact of Efavirenz on Aflatoxin B1 Metabolism: Insights from a Physiologically-Based Pharmacokinetic Model and a Human Liver Microsome Study" is a well-structured paper that provides valuable insights into the interactions between aflatoxin B1 (AFB1) and efavirenz (EFV), an important topic given the prevalence of both substances in certain populations.
1. The manuscript does not explain the rationale behind the chosen doses of Efavirenz (EFV) and Aflatoxin B1 (AFB1). It is recommended that the authors provide a justification for these specific doses, detailing how they relate to therapeutic or environmental exposure levels, or their relevance in the model systems used. Simplification of Figures and Tables:
2. Some figures and tables need to be revise. For instance, Figure 1, which illustrates the human metabolic pathways of AFB1, is cluttered and some labels are too small to read easily. It is suggested to use larger fonts, minimize non-essential information, and highlight the main metabolic pathways of AFB1. Indicators should clearly distinguish between less toxic and more toxic metabolites for better reader comprehension. In Table 1, the dense presentation of pharmacokinetic parameters makes it difficult to distinguish between different treatment groups. Using color coding or different formatting to highlight key comparative data could help readers quickly identify essential information.
3. While the use of statistical methods is mentioned, specific tests and the significance levels (e.g., p-values) of the results are not clearly reported. It is recommended to indicate significance levels for all comparisons directly in the figures and tables to enhance the transparency and reliability of the research findings.
4. The discussion section largely describes results without extensive comparative analysis. The authors should consider expanding their discussion to include comparisons with other studies, addressing any discrepancies found, and exploring possible reasons for these differences. This would not only deepen the discussion but also help readers to more comprehensively understand the significance of the findings.
5. The description of the preparation and handling of human liver microsomes (HLM) is vague. Detailed experimental protocols should be provided to allow for reproducibility and to give readers a clear understanding of the methods employed.
6. I would suggest the authors add a more comprehensive discussion about the implications of these limitations and specific recommendations for future research would be valuable. This could include potential studies that could address unanswered questions or test the findings in other populations or settings.
7. Some studies can by cited for more comprehensive references. For example, a recent review in AFB1 can be cited in the beginning of the introduction https://doi.org/10.1016/j.bcp.2022.115005 .
Reviewer 3 Report
Comments and Suggestions for Authors
The manuscript clearly presents the impact of Efavirenz on Aflatoxin B1 Metabolism using a Physiologically-Based Pharmacokinetic Model 3 and a Human Liver Microsome. The experimental design is appropriate to the presented hypothesis, and the analysis methods used are sufficiently well described to be reproducible.
The data presented are well interpreted and relatively easy to understand. Table 1 should be divided into 4 different tables for easier tracking and understanding of the data presented. The results are interpreted accordingly, but the discussions and conclusions must be supported by more data from the references
Round 2
Reviewer 1 Report
Comments and Suggestions for Authors
Sorry, I do not see the significant improvement of the manuscript. Based on my opinion, the experimental model applied is not proper. Further experiments are required, applying at least in vitro studies on primary human cultured hepatocytes or on liver slices. Therefore, my previous suggestion did not change, and I do not recommend the publication of this work.
Comments on the Quality of English Language-
Author Response
We understand the remarks of reviewer 1 if this would be an induction experiment, but it is not the case here. It is a food-contaminant drug interaction experiment and the state of the art for DDI (drug-drug interaction) is to perform this in microsomes. It was extensively explained in the rebuttal letter but for some reason there is still the proposal to perform these experiments in liver slices....something that is not in the scope of these experiments. Experiments in hepatocytes or liver slices can be performed additionally to have more information on the pharmacokinetics of AFB1 but it has nothing to do with what was performed here...being food-contaminant-drug interactions in combination with PBPK-modelling. We have asked external people on the design of these experiments and they were approved, also reviewer 2 and 3 had no issues at all on the design of this experiment.